# Evaluating Olympic Pictograms Using Fuzzy TOPSIS—Focus on Judo, Taekwondo, Boxing, and Wrestling

**DOI:** 10.3390/ijerph19073934

**Published:** 2022-03-25

**Authors:** Kyoungho Choi, Bongseok Kim, Jinhee Choi

**Affiliations:** 1Department of Radiological Science, Research Institute of Health Statistics, Jeonju University, Jeonju-si 55069, Korea; ckh414@jj.ac.kr; 2Department of Sports Coaching, Jeonju University, Jeonju-si 55069, Korea; kkkbbbq@jj.ac.kr; 3Department of Fashion Business, Jeonju University, Jeonju-si 55069, Korea

**Keywords:** pictogram, TOPSIS, sensitivity analysis, decision-making, weight

## Abstract

It is necessary to evaluate whether Olympic pictograms are designed accurately and are easy to understand, so that they fulfill their intended functions and roles. Olympic pictograms are used to facilitate smooth communication at this large sporting event. However, viewers often find it challenging to understand the actual sport represented by the pictogram. This study evaluates the ranking of comprehensibility of the pictograms for judo, taekwondo, boxing, and wrestling used in six games, from the 27th Sydney Olympics in 2000 to the 32nd Tokyo Olympics in 2021. The evaluation was done using the fuzzy technique for order of preference by similarity to ideal solution (TOPSIS) method, a multi-criteria decision-making methodology commonly used in economics and other fields. Data collection was conducted from 10 May to 30 June 2021 for 44 general public and seven experts. The results are as follows. First, the pictograms from the 2008 Beijing Olympics ranked first in three sports: taekwondo, boxing, and wrestling, but there were no pictograms that consistently ranked first or sixth in all sports. Second, the sensitivity analysis result shows the possibility that the ranking would be reversed if the weight of the evaluation factors were changed. This study is expected to contribute to developing pictograms that can adequately convey the appropriate information regarding Olympic sports in the future.

## 1. Introduction

The Olympics is an international event wherein thousands of athletes compete in several sporting events. Moreover, the Olympics are held every two years, alternating between the Summer Olympics and Winter Olympics, under the aegis of the International Olympic Committee (IOC). The games originated from the ancient Greek games held in Olympia, Greece, from the 8th century BCE to the 5th century CE.

Today, the Olympics have developed into an international sports and culture festival that brings people from around the world together to uphold international amity, friendship, peace, and human culture in a pure sporting spirit transcending religion, race, and thought [1].

The attendance of individuals from different countries and races at the Olympics transcends language and culture. To facilitate smooth communication at such a large sporting event, the most representative nonverbal communication is used: Olympic pictograms. They first appeared at the 1936 Berlin Olympics in Germany but became official at the 1964 Tokyo Olympics [2].

A pictogram is a compound image combining a picture and a telegram that presents a message; it is used in international events to facilitate understanding and communication. A pictogram is an icon that has the character of a picture. Shin [3] defines it as “a consistent means of communication in the modern society that requires faster and more accurate communication” and points out that it is achieving rapid spread and development, and the proportion of pictograms is increasing day by day. An icon is a symbol that uses a characteristic part of an object that is replaced by the object itself, whereas a symbol is in a fundamentally arbitrary relationship with the represented object [4]. According to Bernard and Marcel [5], symbols are related to meaning, whereas pictograms are related to the degree of description.

The pictogram has several functions: to show the final output of a process, to allow something, to warn against something, and to forbid something [6]. It should have the following characteristics: easy identification, clear visualization, accessibility and legibility, and clear graphic representation. Because pictograms are used during the Olympic events, they must be visually acceptable to people from various cultural backgrounds; they must also be understood easily, quickly, and accurately by the general public.

Representative studies related to Olympic pictograms include those by Kim [7], who can be seen as a pioneer in related studies; a study on emphasizing the function as a graphic language by Adir et al. [6]; a study on the process of making pictograms for the Summer Olympics, also by Adir. et al. [8]; and a study on pictograms and accessibility of Olympic and Paralympic by Akiyama [9]. However, considering the usage and importance of Olympic pictograms and the fact that many people are using them, there are relatively few related studies. Moreover, the standards and forms of pictograms used to express sports activities per the cultural characteristics of the host country are often unclear. As a result, viewers often find it challenging to understand the actual sport represented in the pictogram. To avoid such inconvenience, the Olympic pictograms must satisfy the general conditions of universality, interest, and suitability.

However, few studies have evaluated whether Olympic pictograms convey information accurately to people. Therefore, it is necessary to evaluate whether the pictograms related to Olympic sports enjoyed by people around the world are made to be accurately understood by faithfully performing their functions and roles. To this end, this study used the Fuzzy technique of order preference by similarity to ideal solution (TOPSIS), a multi-criteria decision-making methodology (MCDM). The MCDM is an analysis method suitable for decision-making by selecting optimal alternatives based on a number of criteria and alternatives and was considered to be the most suitable for achieving the purpose of this study.

Through the Fuzzy TOPSIS method, an MCDM, this study evaluated the degree of understanding the pictograms of judo, taekwondo, boxing, and wrestling that was used in the six Olympic games from the 27th Sydney Olympics in 2000 to the 32nd Tokyo Olympics in 2021. The reason for selecting four combat sports among the Olympic sports was that they were considered appropriate for evaluating the degree of understanding the pictograms as they were not popular compared to ball games such as soccer and baseball. The target period chosen began with the Sydney Olympics because Taekwondo was adopted as an official sport from then on and Taekwondo pictograms were used for publicity and information.

The results of this study will be helpful in developing pictograms that can appropriately convey information about future Olympic sports. Consequently, the results can be utilized and applied to the development of pictograms for other sports, thereby contributing to the enjoyment of Olympic games by more people around the world.

This study is organized as follows. Section 1 describes the definition and role of the pictogram, as well as the prior studies and purpose of the Olympic pictograms. Section 2 deals with a theoretical overview of the fuzzy technique for order of preference by similarity to the ideal solution (TOPSIS), and Section 3 considers research methodologies, including determining evaluation criteria, collecting data, and explaining analysis tools. Section 4 conducts a priority evaluation and sensitivity analysis using the fuzzy TOPSIS, and Section 5 and Section 6 present the results and conclusions, respectively.

## 2. Materials and Methods

This study utilizes the fuzzy TOPSIS, a multi-criteria decision-making methodology (MCDM), to rank the comprehensibility of pictograms for judo, taekwondo, boxing, and wrestling in the six Olympic games from the 27th Sydney Olympics in 2000 (when taekwondo was accepted as an official sport) to the 32nd Tokyo Olympics in 2021. The findings of this study will help develop pictograms that can adequately convey information regarding Olympic sports in the future. They can also be used to develop pictograms in other sporting events, enabling more people around the world to enjoy Olympic sports.

The main research problems of this study are as follows.

Research Question 1: To evaluate and prioritize the comprehensibility of the Olympic pictograms for judo, taekwondo, boxing, and wrestling events using the fuzzy TOPSIS method.Research Question 2: A sensitivity analysis is to be conducted on the TOPSIS results to explore the possibility of a change in priority resulting from weight changes.

### Theoretical Reflection on the Methodology of Fuzzy TOPSIS

Hwang and Yoon [10] proposed the TOPSIS methodology based on the linear ordering method proposed by Hellwig in 1968. It was expanded into the fuzzy TOPSIS by Chen [11] using triangular fuzzy numbers, as shown in Figure 1. The theoretical background for the fuzzy TOPSIS is well summarized in Salabun [12] and Dudek and Jefmanski [13]. A summary of the fuzzy TOPSIS is provided in this section.

Figure 2 presents the hierarchy of this study’s decision-making problem. Table 1 shows the decision-making matrix, composed of $n^{}$ alternatives $A_{1},\;A_{2},\;\cdots ,\;A_{n}$ and the $m^{}$ evaluation criteria $C_{1},\;C_{2},\;\cdots ,\;C_{m}$. Here, $\tilde{x_{ij}}$ is the fuzzy value, and $\tilde{w_{j}}$ is the fuzzy weight of $m^{}$ evaluation criteria.
(1)$$\mu_{A}=\{\begin{array}{c} 0\;\;\;(x<a_{1\;})\;\;\\ \frac{\;\;x-a_{1}}{a_{2}-a_{1}}\;\;\;\left(a_{1}\;\leq x\;\leq \;a_{2}\right)\\ \;\frac{a_{3}-x}{a_{3}-a_{2}}\;\;\;\left(a_{2\;}\leq x\;\leq \;a_{3}\right)\\ 0\;\;\left(x\;\geq \;a_{3}\right) \end{array}$$



$$W=(w_{1},\;w_{2},\;\cdots ,\;w_{m})$$



The process of selecting the nearest alternatives to the fuzzy positive ideal solution (FPIS) and fuzzy negative ideal solution (FNIS) based on the principle of TOPSIS can be divided into the following seven stages.

First stage: Calculate $\tilde{x_{ij}}$, the fuzzy value of the constitutive element of the decision-making matrix in Table 1 from the sample population.

Second stage: Calculate $\tilde{w_{j}}$, the fuzzy weight of the evaluation criteria from the respondents consisting of experts.

Third stage: Calculate the fuzzy weighted normalized matrix by normalizing $\tilde{x_{ij}}$ calculated in the first stage.
(2)$$\tilde{z_{ij}}=\frac{\tilde{x_{ij}}}{\sqrt[{}]{\sum_{i=1}^{n}\tilde{x_{ij}^{2}}}},\;i=1,\;\ldots \ldots ,\;n;j=1,\;\ldots \ldots ,\;m.$$

Fourth stage: Calculate the weighted and normalized fuzzy value.
(3)$$\tilde{v_{ij}}=\tilde{w_{ij}\;}\;\tilde{z_{ij}}$$

Fifth stage: Calculate FPIS and FNIS. Here, $J_{1}$ and $J_{2}$ implies the evaluation criteria for benefits and costs, respectively.
(4)$$\tilde{A^{+}}=\left\{\tilde{v_{1}^{+}}\;,\;\tilde{v_{2}^{+}}\;,\;\ldots ,\tilde{v_{m}^{+}}\right\}=\{(\;max_{i}\;\tilde{v_{ij}}\;|\;j\in \;J_{1}),\;(min_{i}\;\tilde{v_{ij}}\;|\;j\in \;J_{2})\;|\;i=1,\ldots \ldots ,n\}\;\tilde{A^{-}}=\left\{\tilde{v_{1}^{-}}\;,\;\tilde{v_{2}^{-}}\;,\;\ldots ,\tilde{v_{m}^{-}}\right\}=\{(\;min_{i}\;\tilde{v_{ij}}\;|\;j\in \;J_{1}),\;(max_{i}\;\tilde{v_{ij}}\;|\;j\in \;J_{2})\;|\;i=1,\ldots \ldots ,n\}\;$$

Sixth stage: Calculate $d_{i}^{+}$, the Euclidean distance between each value at the fuzzy weighted and normalized matrix and FPIS as well as $d_{i}^{-}$, the Euclidean distance between each value and FNIS.

Seventh stage: Finally, find alternatives closest to the ideal solution and furthest from the negative solution. $CC_{i}^{+}$ has a value between 0 and 1. The alternative with the greatest value is the optimal idea.
(5)$$CC_{i}^{+}=\;\frac{d_{i}^{-}}{d_{i}^{+}+d_{i}^{-}}\;,\;\;i=\left(1,\ldots \ldots ,\;n\right)$$

## 3. Research Methodology

### 3.1. Research Flow Chart

To evaluate alternatives using fuzzy TOPSIS, it is necessary to first set alternatives and then determine evaluation criteria. After that, it is necessary to collect data and prioritize them. The flow chart of the process of this study is presented in Figure 3.

### 3.2. Setting Alternatives

Establishing an alternative is the first task in designing a multi-criteria decision-making problem. Alternatives refer to the target of evaluation, which must be clear, actionable, and available for evaluation. In this study, the Olympic pictograms for judo, taekwondo, boxing, and wrestling presented in Table 2 were selected as alternatives for comprehensibility evaluation.

### 3.3. Determining the Evaluative Criteria

In a multi-criteria decision-making problem, the evaluation criteria refer to the ones used in evaluating alternatives. Several evaluation criteria may be considered in the decision-making problem, but not all can be used. Furthermore, there is no absolute method of setting the evaluation criteria, but general criteria must be configured not to overlap or be skipped.

The evaluation criteria of this study were set as shown in Table 3, combining the general conditions that pictograms should possess (e.g., the function of delivering meaning easily and quickly) and the requirement of an infographic that delivers information using graphics [14]. According to Pettersson [15] and Noh and Son [16], “infographic” is a compound word combining information and graphics as a method of information design that transmits verbal and visual messages through various media.

### 3.4. Data Collection

The value corresponding to each element of the decision-making matrix consisting of alternatives and evaluation criteria is the evaluation value of the alternative. This study thus conducts sample selection by quarter random sampling the general public watching the Olympics stratified by sex and age. The survey was conducted from 10 May to 30 June 2021, wherein, through a self-enumeration method, participants were asked to respond. Because this study is not a survey aimed at a statistical test, type I and type II errors are not considered. Therefore, calculating the sample size has minimal significance. In the case of fuzzy TOPSIS, the sample size was not determined in most previous studies and is generally not large, as in the study by Kabir [17], which consisted of seven people, and the study by Percin [18], which consisted of 14 people. However, in this study, our survey comprised 44 people; as mentioned earlier, we considered the characteristics of sex and age. Meanwhile, the expert survey for determining the weight that indicates the importance of the five selected evaluation criteria, shown in Table 3, was conducted on seven experts. The selection criteria included being university professors with a doctorate degree or over three years of experience working in relevant companies and domains. Based on these criteria,—three professors from the fashion design department (Ph.D., fashion designer, fashion merchandiser), two professors from the sports coaching department (Ph.D., taekwondo player, soccer player) and two professors from the industrial design department (Master’s degree, furniture designer) were selected.

The survey to evaluate the level of understanding and to determine weight used the 5-scale linguistic rating, as shown in Table 4.

Next, the measured values obtained by verbal evaluation were converted into fuzzy numbers according to Kore et al. [19] and presented in the range of 1 to 9 points, as shown in Table 5.

### 3.5. Analytical Tools

This study utilizes two analytical tools. First, the “FuzzyTOPSISLiner” function from the “FuzzyMCDM” package in the R-program by Dudek and Jefmanski [13]; Comprehensive R Archive Network [18] was used to resolve Research Question 1. Second, the PyTOPS program developed by Yadav et al. [20] in Python was used for the sensitivity analysis in the TOPSIS for Research Question 2.

### 3.6. Ethical Considerations

Because data collection involved human subjects, the study was first reviewed by Jeonju University’s institutional review board (IRB) (jjRB-210413-HR-2021-0413). Coffee coupons were provided to respondents to express gratitude and facilitate an environment where they could provide honest responses.

## 4. Results

### 4.1. Evaluation of Priority Ranking Using Fuzzy TOPSIS

The responses collected from seven experts to determine the weight of the five evaluation criteria presented in Table 2 were integrated according to the fuzzy response synthesis algorithm presented in Kore et al. [21]. The major results are as follows.

First, the fuzzy weight W for the five evaluation criteria in Table 2 is as follows. W= (7, 9, 9), (1, 5.857, 9), (1, 2.714, 7), (3, 7, 9), (3, 7, 9). The sex distribution of the respondents was 52.3% male and 47.7% female, and 34.1% were in their 20s or younger, 20.5% in their 40s, and 13.6% in their 50s or over. A summary of the integrated fuzzy responses focusing on $a_{2}$, obtained from the respondents, is presented in Table 6.

Table 7 presents the resulting priority ranking of the six alternatives by sports following the procedure of selecting the nearest alternatives to the FPIS and FNIS based on the principle of TOPSIS. The “FuzzyTOPSISLiner” function of the “FuzzyMCDM” package in the R-program was used to analyze the data in Table 5. No event consistently ranked first or sixth across all sports. However, the 2008 Beijing Olympics pictograms ranked first for three sports, including taekwondo, boxing, and wrestling.

### 4.2. Sensitivity Analysis

A sensitivity analysis was performed to ascertain the result of TOPSIS on the comprehensibility of the Olympic pictograms using Yadav et al.’s [20] PyTOPS program developed in Python. This was done to examine the possibility of a change in priority dependent on a change in the weights in the priority evaluation in the TOPSIS method. In other words, the study’s findings are robust if the priority does not change significantly even when the weights in the five evaluation criteria change. However, if the change in priority is significant even with a small change in the weights, indicating a high sensitivity of the measurement results, the results are not robust.

For sensitivity analysis, the changes in weights were 25%, 40%, and 55%. Within the given weight change, the simulation was performed 1,000 times; the result is presented in Table 8. It shows the means and standard deviations of the closeness to the ideal solution of top alternatives according to weight changes. Figure 4 is an example of judo, and it can be seen that the higher the ranking, the closer the average is to the ideal solution, even if the weights change.

## 5. Discussion

The Olympics is not just a sporting event; it has developed into a grand festival where all participating nations come together as a community comprising different religions and cultures. Furthermore, the Olympic Games provide a prime opportunity for the host countries to promote their culture and traditions to the world; it also offers other economic profit-generating and national development opportunities. In other words, hosting the Olympics is a step forward for many countries.

Hence, the host country usually strives to promote and communicate the Olympics sports to the world in various ways. A typical method is by using a pictogram; this started with the 1964 Tokyo Olympics. Ease of communication is the most important function of pictograms in the Olympics. Thus, it is undesirable to have pictograms representing specific events that are difficult to understand due to cultural characteristics unique to the host country. Therefore, Olympic pictograms should satisfy the general conditions of pictograms, such as universality, interest, and suitability.

The Olympic flag, as a symbol of the Olympics, is made up of blue, black, yellow, green, and red rings, and each color represents a continent: Europe in blue, Africa in black, Asia in yellow, Oceania in green, and America in red [22]. As most countries in the world participate in the Olympics, there are many difficulties in communication, including language differences. To solve this problem, pictograms were introduced in the 1964 Tokyo Summer Olympics. The pictograms are also called pictorial symbols because they have the characteristics of a picture among icons [23]. While using pictograms to promote or guide the Olympics is varied, one of the most important reasons is to convey information quickly and easily to linguistically and culturally diverse people. These pictograms should be easy to visibly understand [15], deliver information rapidly, emphasize symbolic signs visually, contain aesthetic features, and be easy to understand without prior training. Among these, one of the greatest advantages of pictograms is readability; anyone can easily understand their meaning.

However, in the major studies on the interpretation of the Olympic pictograms, they are often interpreted through the subjectivity of the researchers using the symbolic sculptures, animals, and colors of the host country. For example, Park [1] interpreted them through the subjectivity of researchers, using symbolic sculptures, animals, and colors that represent the host country. The 2008 Beijing Olympics Pictograms were produced through a fusion of ancient Chinese pictographs and simple features of modern shapes based on handwriting to convey the ideology of the Olympics and the cultural significance of the host country [24]. As such, the production and design of Olympic pictograms do not deviate from the subjective interpretation of researchers. Therefore, beyond those studies, this study analyzed the effectiveness, suitability, universality, and appeal of Olympic pictogram designs and determined that the Olympic spirit, which pursues universal values such as the development of humanity through participation [25], can be passed on to the general public, Olympic athletes, media officials of each country, and the Olympic organizers through these designs.

Based on the above, Olympic pictograms should be made to satisfy general conditions that must be met from a design perspective, such as universality, interest, and suitability. To identify improvements and develop better alternatives, it is necessary to evaluate pictograms from an academic perspective and identify specific necessary characteristics based on the results. Unfortunately, it is difficult to find studies evaluating the Olympic pictograms from a design perspective

The comprehensibility ranking of the pictograms for judo, taekwondo, boxing, and wrestling used in the six games from the 27th Sydney Olympics in 2000 to the 32nd Tokyo Olympics in 2021 was evaluated. The evaluation was done using the fuzzy TOPSIS method, a multi-criteria decision-making methodology commonly used in economics and other fields. The results are as follows.

First, the pictograms from the 2008 Beijing Olympics ranked at the top in three sports: taekwondo, boxing, and wrestling, but there were no pictograms that consistently ranked first or sixth in all sports. This is presumed to be due to the simplicity of the Beijing Olympic pictograms. Overall, no pictograms were consistently evaluated as the first or sixth place in all sports, but the pictograms, which were evaluated in sixth place, showed low clarity. The variety of evaluation results is similar to the results of Yang [26], which evaluated the logo design of university hospitals.

Second, the result of the sensitivity analysis shows a possibility that the ranking will be reversed if the weight of the evaluation factors changes; however, in the 1000-time repetitive prediction, the better the evaluation ranking, the closer the value of the priority ranking to the ideal solution on average even if the weight changes. Paradoxically, such a result implies that incorrect weighting of the evaluation factors may lead to an incorrect evaluation, suggesting the importance of the former. Furthermore, each evaluation factor can significantly affect the evaluation results.

Finally, this study has some limitations. It only targeted four combat sports and used five evaluation criteria for the purposes of the study. Therefore, the results of this study should not be interpreted broadly as an evaluation of the pictograms of the remaining sports of the six Olympic Games to be analyzed.

## 6. Conclusions

Pictograms were developed as a means of communication before writing systems were established, and the paintings engraved on prehistoric caves and sculptures fall into this category. Some pictograms have been in use since 1900 and have retained their meaning, and the IOC first used pictograms at the Tokyo Olympics in 1964. In recent years, the use of pictograms has increased, and they are used as announcements and warnings. This is because pictograms are visual languages in the form of pictographs produced with simple forms and content that attract attention. To promote pictograms at the Olympics, we need to understand their use objectively to determine what improvements are necessary.

This study considered five evaluation factors: clarity, familiarity, entertainment, attractiveness, and identity, based on the conditions that infographics and pictograms must satisfy. However, in evaluating the Olympic pictogram, there may be better factors other than the five evaluation factors set out in this study; thus, further research is needed in this area. This study is significant in that it raises the need for evaluation of whether the Olympic pictogram is performing its function and role and suggests an evaluation method. Additionally, from a practical point of view, this study is expected to contribute to the development of pictograms as a means of promoting Olympic sports in the future.

## Figures and Tables

**Figure 1 ijerph-19-03934-f001:**
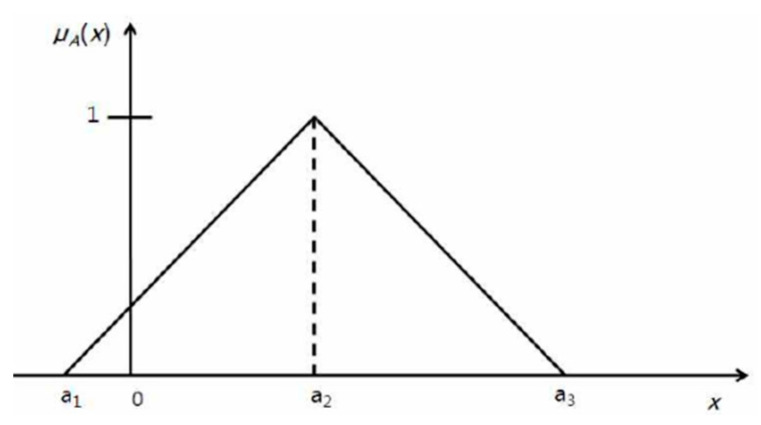
Triangular fuzzy numbers. A triangular fuzzy number is a fuzzy number represented with three points, as follows: A = ($a_{1},\;a_{2},a_{3}).$ This representation is interpreted as membership functions.

**Figure 2 ijerph-19-03934-f002:**
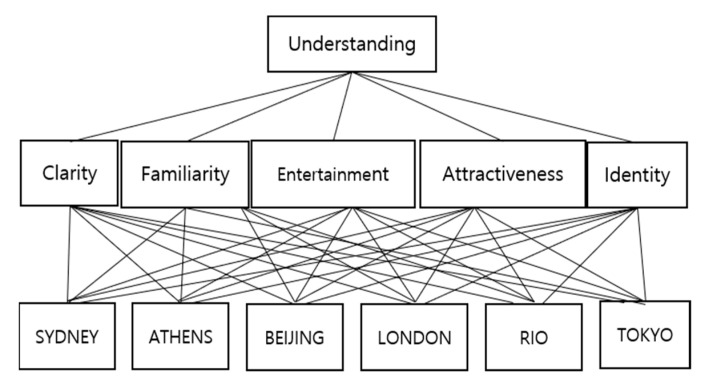
Hierarchy of the decision-making problem.

**Figure 3 ijerph-19-03934-f003:**
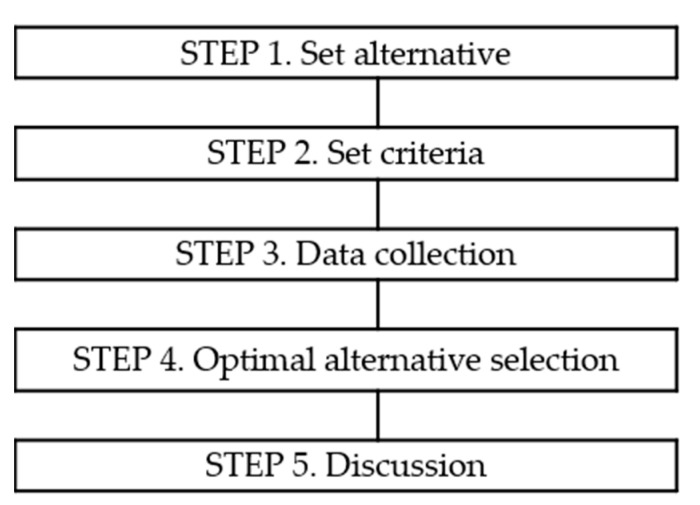
Process of this study.

**Figure 4 ijerph-19-03934-f004:**
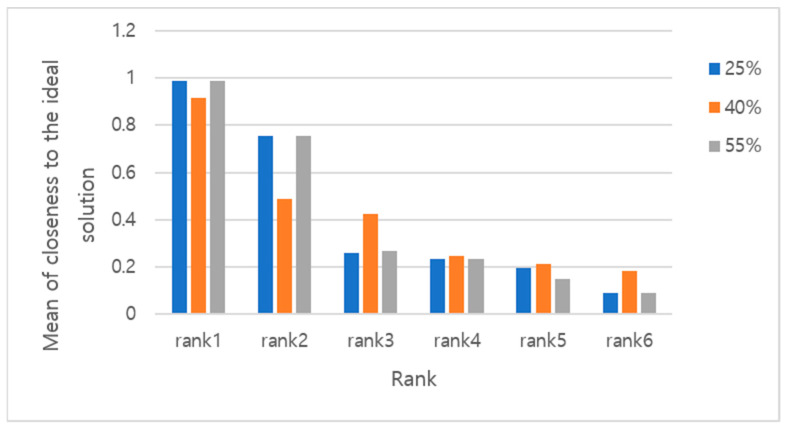
Means of the closeness to the ideal solution of top alternatives (exam: Judo).

**Table 1 ijerph-19-03934-t001:** Decision-making matrix. The columns show the criteria, and the rows list the alternatives.

	C1 Clarity	C2 Clarity	C13 Clarity	C4 Clarity	C5 Clarity
$A_{1}$ SYDNEY	$\tilde{x_{11}}$	$\tilde{x_{12}}$	$\tilde{x_{13}}$	$\tilde{x_{14}}$	$\tilde{x_{15}}$
$\;A_{2}$ ATHENS	$\tilde{x_{21}}$	$\tilde{x_{21}}$	$\tilde{x_{23}}$	$\tilde{x_{24}}$	$\tilde{x_{25}}$
$\;A_{3}$ BEIJING	$\tilde{x_{31}}$	$\tilde{x_{32}}$	$\tilde{x_{33}}$	$\tilde{x_{34}}$	$\tilde{x_{35}}$
$\;A_{4}$ LONDON	$\tilde{x_{41}}$	$\tilde{x_{33}}$	$\tilde{x_{43}}$	$\tilde{x_{44}}$	$\tilde{x_{45}}$
$\;A_{5}$ RIO	$\tilde{x_{51}}$	$\tilde{x_{34}}$	$\tilde{x_{53}}$	$\tilde{x_{54}}$	$\tilde{x_{55}}$
$\;A_{6}$ TOKYO	$\tilde{x_{61}}$	$\tilde{x_{35}}$	$\tilde{x_{63}}$	$\tilde{x_{64}}$	$\tilde{x_{65}}$

**Table 2 ijerph-19-03934-t002:** Pictogram of Judo, Boxing, Wrestling, Taekwondo at Olympic Games (2000–2020). The columns show the year, and the rows list the criteria.

	2000	2004	2008	2012	2016	2020
Judo	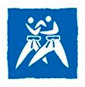	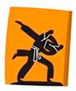	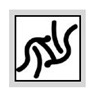	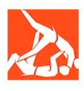	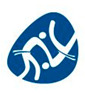	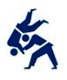
Boxing	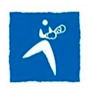	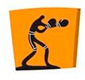	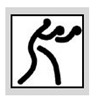	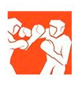	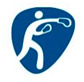	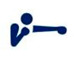
Wrestling	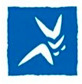	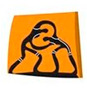	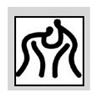	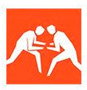	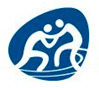	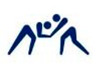
Taekwondo	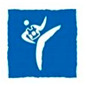	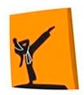	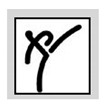	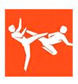	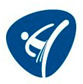	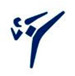

**Table 3 ijerph-19-03934-t003:** Evaluation Criteria for Comprehensibility of the Olympic Pictograms.

Evaluation Criterion	Meaning
Clarity	The content of the information shall be clear, and easily and quickly understandable.
Familiarity	It shall visually stimulate curiosity and not be unpleasant.
Entertainment	It shall be humorous and fun to approach.
Attractiveness	It shall give a sense of satisfaction that enables a continuous usage of visual information.
Identity	It shall reflect the culture and tradition of the host country.

**Table 4 ijerph-19-03934-t004:** Linguistic ratings and weight for the alternatives.

Linguistic Ratings	Weight
Very poor	Very low
Poor	Low
Fair	Medium
Good	High
Very good	Very high

**Table 5 ijerph-19-03934-t005:** Fuzzy ratings for linguistic rating.

Fuzzy Number	Linguistic Ratings	Weight
(1, 1, 3)	Very poor	Very low
(1, 3, 5)	Poor	Low
(3, 5, 7)	Fair	Medium
(5, 7, 9)	Good	High
(7, 9, 9)	Very good	Very high

**Table 6 ijerph-19-03934-t006:** Integrated fuzzy responses. The columns list the criteria and the rows show the alternatives.

		Clarity	Familiarity	Entertainment	Attractiveness	Identity
Judo	Sydney	4818	4954	5136	4863	5227
	Athens	7636	7090	6045	6500	5818
	Beijing	5590	5090	5363	4772	5545
	London	5727	5863	5954	5090	4772
	Rio	5272	5090	5545	5590	4954
	Tokyo	7590	7090	6409	7000	7090
Taekwondo	Sydney	6591	6500	5136	5227	5273
	Athens	8045	7682	5409	6773	6636
	Beijing	4864	5091	5773	4909	5182
	London	5818	5455	6773	6136	5227
	Rio	5591	5318	5818	5318	4090
	Tokyo	5318	5273	4909	4273	5000
Boxing	Sydney	3909	3500	4500	4000	4273
	Athens	6818	5818	5545	5182	5591
	Beijing	5318	5091	5818	4955	5500
	London	8636	8091	6636	7091	7045
	Rio	6773	6182	5864	5864	6000
	Tokyo	6045	5682	5773	5364	5455
Wrestling	Sydney	2272	2500	3681	3318	3227
	Athens	3681	3818	5045	4636	4590
	Beijing	4454	4636	5272	4863	5363
	London	6545	6227	5863	6090	5636
	Rio	6909	6954	6590	6545	5727
	Tokyo	6363	6000	5454	5227	5136

**Table 7 ijerph-19-03934-t007:** Priority ranking of six alternatives by sports. The columns list the criteria and the rows show the ranking.

	Judo	Taekwondo	Boxing	Wrestling
1	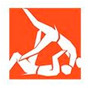	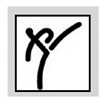	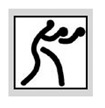	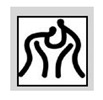
London	Beijing	Beijing	Beijing
2	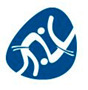	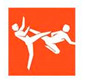	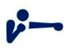	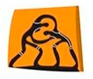
Rio	London	Tokyo	Athens
3	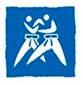	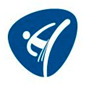	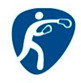	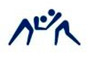
Sydney	Rio	Rio	Tokyo
4	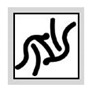	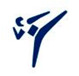	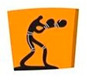	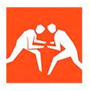
Beijing	Tokyo	Athens	London
5	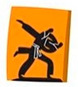	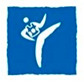	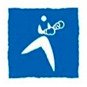	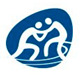
Athens	Sydney	Sydney	Rio
6	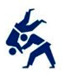	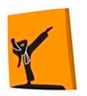	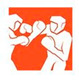	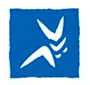
Tokyo	Athens	London	Sydney

**Table 8 ijerph-19-03934-t008:** Means and standard deviations of the closeness to the ideal solution of the top alternatives. The columns list the criteria for each event, and the rows show the ranking.

	Judo			Taekwondo			Boxing			Wrestling		
	25%	40%	55%	25%	40%	55%	25%	40%	55%	25%	40%	55%
1	0.989 (0.002)	0.916 (0.021)	0.989 (0.002)	0.916 (0.013)	0.916 (0.021)	0.916 (0.028)	1.000 (0.000)	1.000 (0.000)	1.000 (0.000)	1.000 (0.000)	1.000 (0.000)	1.000 (0.000)
2	0.755 (0.027)	0.489 (0.012)	0.757 (0.055)	0.486 (0.008)	0.489 (0.012)	0.491 (0.017)	0.602 (0.001)	0.602 (0.002)	0.602 (0.003)	0.886 (0.005)	0.886 (0.009)	0.885 (0.012)
3	0.260 (0.013)	0.423 (0.030)	0.268 (0.023)	0.425 (0.022)	0.423 (0.031)	0.419 (0.037)	0.530 (0.013)	0.529 (0.020)	0.527 (0.027)	0.779 (0.015)	0.779 (0.025)	0.775 (0.034)
4	0.232 (0.010)	0.246 (0.021)	0.231 (0.017)	0.239 (0.013)	0.245 (0.020)	0.253 (0.029)	0.453 (0.001)	0.453 (0.003)	0.454 (0.004)	0.514 (0.011)	0.515 (0.018)	0.516 (0.025)
5	0.195 (0.012)	0.210 (0.024)	0.148 (0.021)	0.212 (0.016)	0.209 (0.023)	0.208 (0.032)	0.335 (0.006)	0.335 (0.010)	0.336 (0.014)	0.351 (0.010)	0.352 (0.016)	0.354 (0.022)
6	0.089 (0.011)	0.181 (0.020)	0.087 (0.023)	0.184 (0.014)	0.179 (0.021)	0.174 (0.026)	0.000 (0.000)	0.000 (0.000)	0.000 (0.000)	0.000 (0.000)	0.000 (0.000)	0.000 (0.000)

## Data Availability

The data presented in the study are available on request from the corresponding authors.

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
