# Peer review of "Evaluating Olympic Pictograms Using Fuzzy TOPSIS—Focus on Judo, Taekwondo, Boxing, and Wrestling"

_ijerph, 2022, doi:10.3390/ijerph19073934_

Round 1

Reviewer 1 Report

Dear authors, I enclose my recommendations:

Abstract
The abstract adequately summarises the research, including the objective, the methodology used, the main findings of the results and the most relevant conclusions. However, the total number of subjects surveyed is not indicated.

Keywords:
I recommend that the keywords are changed to words that do not appear in the title of the paper. This will multiply the search options.

Introduction

  • The concept "icon" is defined but not "pictogram", it must be, it is spoken of.
  • Line 64: Which studies? However, only a few studies have examined the suitability of the Olympic 64 pictograms
  • Indicate section 1-->This study is organized as follows. Section 2 deals with...
  • Line 43: American philosopher Peirce C. E., date?
  1. Materials and Methods

2.1. Subjects

- Gender: is a cultural term. Sex should be used.

Results

- What does this refer to? Isolated phrase. You should not start a sentence like this. W= [(7,9,9), (1,5.857,9), (1,2.714,7), (3,7,9), (3,7,9)].

Discussion

- Line 304: Different font type: Yang [25],

Conclusion

It is recommended to incorporate some practical proposals

Author Response

Thank you for your review and please see the attachment. 

Reviewer 2 Report

I recognize my surprise and quality of the work I have just read. I will recommend that it be published. Thank you very much for such high quality.

I recommend some fresh references about Olympic Games should be included

Author Response

(The authors gave the same response as above.)

Reviewer 3 Report

Dear Authors

You have written an interesting study analysing the pictograms from combat sports from the Sydney Olympic games onwards. However, some parts need to be addressed for greater clarity.

1st paragraph of the Introduction needs referencing. Add

There is no clear rationale why did you chose combat sports and why from the Sydney Olympic games onwards. This needs to be clear. Amend the introduction accordingly.

Important literature also needs to be included:

http://piim.newschool.edu/journal/issues/2012/04/pdfs/ParsonsJournalForInformationMapping_Kim_SoJung.pdf

Also, what does previous research show and which methods have been used to evaluate them? This information also needs to be included in the introduction to be the foundation for your selected methods

Report agreement values and how were they calculated for 7 experts. Also, report inclusion criteria for experts and survey sample.

The limitations of the study paragraph is missing. Add

Overall I recommend major revision.

Kind regards

Author Response

(The authors gave the same response as above.)

Round 2

Reviewer 3 Report

Dear Authors

Thank you for addressing my comments. However, some of them are not done correctly.

Please move the limitations of the study paragraph to the end of the discussion as it should not be in the conclusion.

Also the criteria for experts - So your criteria were university professors, what about experts from the field of marketing? Please describe the inclusion criteria for experts (is just enough to have a PhD or any relevant marketing experience in years) as you just stated who they were. I hope this is clear enough.

Overall the paper has improved and with small corrections, it is suitable for acceptance.

Kind regards

Author Response

Thank you and please see the attachment.

This manuscript is a resubmission of an earlier submission. The following is a list of the peer review reports and author responses from that submission.

Round 1

Reviewer 1 Report

Dear authors, I enclose my recommendations:

Abstract
The abstract adequately summarises the research, including the objective, the methodology used, the main findings of the results and the most relevant conclusions.

Keywords:
I recommend that the keywords are changed to words that do not appear in the title of the paper. This will multiply the search options. 

Introduction
The general idea of the research is presented with an adequate and updated theoretical framework.
LINE 34- Why is this important in the article? "The Olympics have been held across the world; and in 1988 and 2018, South 34 Korea hosted the 24th Summer Olympics and the 23rd Winter Olympics, respectively".
There are definitions that are not justified with any reference - line 45/53-56/
American philosopher Peirce C. E., date?
This form of writing "is well summarized in [11], [12]" is not appropriate, it should indicate the surnames. Correct it, it has been used in several lines throughout the manuscript.

2. Materials and Methods
2.1. Subjects

Why only these games? used in the six games from the 27th Sydney Olympics in 2000 to 11 the 32nd Tokyo Olympics in 2021.

Gender: is a cultural term. Sex should be used.

Please indicate why 44 subjects were chosen for the sample, why not fewer or more? What sampling methodology did you use to draw this particular number of subjects?

4. Priority and Sensitivity Analysis

Table 5. The word taekwondo is cut off.

Results and discussion
These two sections should be separated.
In addition, in the discussion all statements should have previous research that supports or contradicts the results obtained. This should be reviewed. There is no citation.

Conclusions
Are clear.

References
It is not appropriate to use Wikipedia as a reliable source in a scientific article.
They are generally scarce.
The format does not meet the journal's standards.

Author Response

Thank you for your suggestions and comments. I provide the answers to your concerns. Please see the attachment. 

Reviewer 2 Report

This study evaluated the comprehensibility of the pictograms for different sports and various games, which is a promising study; however, I have some comments to improve this study.

  • Abstract missing the importance of the proposed study and comparison with previous work.
  • Weakly related work, I suggest the author find similar studies related to pictograms, not only the Olympics. 
  • The last paragraph in the introduction should summarize the paper's architecture.
  • I recommend the author to follow the instruction of the IJERPH format; instructions are found on the following link: https://www.mdpi.com/journal/ijerph/instructions 
  • Avoid using this style to reference the figures and table (< Figure>).
  • There is a mathematical equation in figure 3, W= [w....]. It's not clear the purpose of this equation; explain the equation separately from the Figure. 
  • All mathematical equations should be rewritten in clear format.
  • A brief introduction to describe the research methodology in figure 4 is recommended.
  • In section 3.2, please describe the meaning of alternative in the context of the proposed work.
  • In table 7, the diagonal line of the table is not correct, and the Judo is written as Yudo. 
  • In section 4.2, the author talks about a simulation. However, there is no description of the simulation or the software used to acquire the results.
  • In section 5, 4th paragraph line (254), the author presents the result achieved in the study, but there is no explanation about why there are no pictograms ranked first in all sports.

Author Response

Thank you for your suggestions and comments. I provide the answers to your concerns. Please refer to attached file. 

Round 2

Reviewer 1 Report

Dear authors, I enclose my recommendations:

Add which studies (references) to justify it: In the case of fuzzy TOPSIS, the sample size is not determined in most previous studies, 

Replace gender by sex: and small samples are possible. In this study, 44 people were surveyed: 10 people in their 20s, 10 in their 30s, 10 in their 40s, 10 in their 50s and older, to consider the characteristics of each gender and age group.

There is no DISCUSSION section. In the discussion, all statements must have prior research to support or contradict the results obtained. This is vital.

The format does not meet the journal's standards. It is still incorrect.

Author Response

Thank you for your suggestions and comments. Please see the attachment.

Reviewer 2 Report

Accept in present form.

Author Response

Thank you for your suggestions and comments. Please see the revised manuscript.

Round 3

Reviewer 1 Report

DISCUSSION
- The discussion should confront previous research with the research presented; in different paragraphs this does not happen.
- This section is not for introducing issues or postulating theories. Everything presented should be constratable.

OVERALL ASPECTS
- Changes in another colour are not marked.
- There is no line numbering to indicate the changes made.
- Shallow scientific depth.